# Prioritising the sources of pollution in European cities: do air quality modelling applications provide consistent responses?

Bart Degraeuwe, Enrico Pisoni, Philippe Thunis

European Commission, Joint Research Centre (JRC), Ispra, Italy

*Correspondence to*: E. Pisoni (enrico.pisoni@ec.europa.eu)

**Abstract.** To take decisions on how to improve air quality, it is useful to perform a source allocation study that identifies the main sources of pollution for the area of interest. Often source allocation is performed with a Chemical Transport Model (CTM) but unfortunately, even if accurate, this technique is time consuming and complex. Comparing the results of different CTMs to assess the uncertainty on source allocation results is even more difficult. In this work, we compare the source allocation (for PM2.5 yearly averages) in 150 major cities in Europe, based on the results of two CTMs (CHIMERE and EMEP), approximated with the SHERPA (Screening for High Emission Reduction Potential on Air) approach. Although contradictory results occur in some cities, the source allocation results obtained with the two SHERPA simplified models lead to similar results in most cases, even though the two CTMs use different input data and configurations.

## 1. Introduction

Air quality models are useful tools to perform a variety of tasks like assessment (simulating concentrations fields at a given moment), forecasting (predicting future concentrations) and source allocation/planning (evaluating priorities of interventions, and the impact of potential emission reduction policies on concentrations). For assessment (Alvaro Gomez-Losada et al., 2018) and forecasting (Corani et al., 2016), it is possible to compare the model results with observations. For example, FAIRMODE[1] (the Forum for air quality modelling in Europe) proposes methods as the Model Quality Indicator and Model Quality Objective (Pernigotti el al., 2013b; Viaene et al., 2016) to assess the quality of the model results for a given application. However, there is no benchmark against which to compare model results for source allocation and planning, as no measurements are available to test the impact of theoretical emission reduction scenarios on concentrations. So, even if very useful to evaluate ex-ante the impact of possible policy options, it is hard to judge the quality of these results. On the other hand, the uncertainty associated to source allocation results can be assessed by comparing them with results from other air quality models (Thunis et al., 2007; Cuvelier et al., 2010; Pernigotti et al., 2013). Both the absolute and relative impacts of emission reductions can then be compared.

---

[1] The Forum for Air quality Modeling (FAIRMODE) was launched in 2007 as a joint response initiative of the European Environment Agency (EEA) and the European Commission Joint Research Centre (JRC). The forum is currently chaired by the Joint Research Centre. Its aim is to bring together air quality modelers and users in order to promote and support the harmonized use of models by EU Member States, with emphasis on model application under the European Air Quality Directives. For more details, see https://fairmode.jrc.ec.europa.eu/.

As an initial phase to design an air quality plan, one is interested in identifying the main sources over a given domain
that are responsible for the pollution at a given location (Isakov et al., 2017). This step is defined in literature as source
allocation (Thunis et al., 2019), i.e. a technique applied to understand the key contributors to air pollution at a given
location. Source allocation then serves as the corner stone to choose the target sector or geographical area when
designing measures for an air quality plan.
The ideal to perform source allocation would be to use directly a Chemical Transport Model (CTM) but this technique
is unfortunately too time consuming to differentiate the impacts of many sources at the same time for various cities in
Europe. An alternative is to simplify the CTM with a so-called source-receptor relationships (SRR) approach, that
mimics the CTM relationships between emission and concentration changes. The most precise SRR would consist in
an independent grid cell-to-grid cell approach. While this approach would allow a high level of flexibility in defining
the zones over which emissions are spatially reduced, it involves simulating independently the effect of emissions
changes in each single grid cell that has pollutant emissions in the model domain. It would require changing precursor
emissions in individual grid cells one at a time and looking at the resulting change in concentrations in each receptor
cell. While theoretically very simple, the resulting number of unknown parameters describing the transfers between
source and receptor cells that need to be identified is very large. For example, for a domain with $50 \times 50$ grid cells
(Ngrid=2500) and 5 precursors (Nprec = 5), the identification of a maximum of 12,500 parameters would be required
(if emissions occur in, and concentration changes need to be calculated for, all grid cells in the domain) to calculate
the change of concentration at a given receptor cell. Therefore 12,500 equations, each connecting concentration
changes and emission changes are necessary to identify these 12,500 unknown parameters. Because each of these
equations requires an independent CTM run, this independent grid cell-to-grid cell option is very costly, and
simplifying assumptions that reduce the number of CTM runs are required (Clappier et al., 2015).
In GAINS ("Greenhouse gas - Air pollution Interactions and Synergies", Amann et al., 2011) the grid-cell to grid-cell
relation is simplified by aggregating source cells into countries. The number of unknown parameters that need to be
identified for one receptor cell equals the number of countries (Ncountry) multiplied by the number of precursors.
This system can only be solved if at least "Nprec x Ncountry" equations are available, requiring a similar number of
independent CTM scenarios. In GAINS, about 50 countries and 5 precursors lead to the need of 250 independent CTM
scenarios to identify 250 unknowns. However, because they are derived from emission reductions at country level,
these SRRs are not applicable at the urban scale.
In the RIAT + tool ("Regional Integrated Assessment Tool", Carnevale et al., 2014). Emissions are aggregated into
'quadrants' that are defined relatively to each grid cell within the domain. The 'quadrant' emissions and their related
grid cell concentrations are then used to feed a neural network that delivers the SRR (Carnevale et al., 2009). Although
the approach requires a limited number of full CTM simulations (around 20), the set-up of the SRR remains complex
due to the need of implementing sophisticated neural networks.
In SHERPA (Thunis et al., 2016; Pisoni et al., 2017), a different approach is taken that reproduces the grid cell-to-
grid cell approach but does not require anywhere near as many CTM runs. SHERPA assumes that the unknown
parameters vary on a cell-by-cell basis but are no longer independent of each other. Instead, these coefficients are
assumed to be related through a bell shape function. With the SHERPA approach, the number of unknown parameters
is then equal to 2 for each precursor and receptor cell. Consequently, for the five precursors of PM2.5 (VOC, $SO_2$,
$NO_x$, PPM and $NH_3$), ten independent CTM simulations are needed for a given receptor cell. Provided that they deliver
independent information, the same CTM scenarios can be used to identify both parameters for all cells within the
domain (see details in Pisoni et al. 2017). Based on these 10 CTM simulations the SHERPA approach allows to quickly
assess the impact of emission reductions for many combinations of sectors, geographical areas and precursors. It is
currently the only approach that allows performing a systematic analysis for about 150 EU cities in terms of sectors
and precursors.
First, the SHERPA SRR approximation of the two CTMs, CHIMERE and EMEP, is built. With these two SRR models
the contribution of 100 sector-area-precursor combinations to the concentration in the city centre is determined and
we assess the similarities and differences between these two set of results. Obviously some of the differences are
caused by the fact that the two CTM models rely on different formulations and parametrisations but also by the fact
that they use different input data (emissions, meteorology…). The objective of this work is to assess the overall
uncertainty (or better, variability) attached to source allocation rather than to assess the sensitivity of the results to a
given parameter (e.g. emissions).
The focus of this study is on PM2.5 yearly averages, because this is the pollutant with the highest impact on human
health, and is therefore a key focus for policy makers in Europe. Because a large number of sources contribute to
PM2.5 concentrations at one location, this is also the most challenging pollutant to manage in air quality plans.
The paper is structured as follows. We briefly present the two Chemical Transport Model and their set-up in Section
2. We then describe the SHERPA methodology and its assumptions in Section 3. Section 4 details the methodology
followed for the source allocation, while the inter-comparison of the results is presented in Section 5. Conclusions are
proposed in Section 6.

## 2.    CHIMERE and EMEP Chemical Transport Models: set-up and simulations

In this work, we use two set of model simulations, performed with two of the leading chemical transport models in
Europe: CHIMERE and EMEP. More details on the models can be found in Mailler et al., 2017 and Couvidat et al.,
2018 (for CHIMERE) and Simpson et al., 2012 (for EMEP). Because a brute force source allocation for 150 cities
with these models would be too time consuming, we use two sets of SHERPA Source Receptor Relationships (SRR),
each based on a training set of about 20 CHIMERE and EMEP CTM simulations . These SRR are then used to perform
the source allocation. Details on the SHERPA training for CHIMERE can be found in Clappier et al., 2015, and for
EMEP in Pisoni et al., 2019.
The CHIMERE and EMEP modelling set-up differ in the following aspects:
• Grid setting: CHIMERE uses a grid of 0.125 degrees longitude by 0.0625 degrees latitude, corresponding to
rectangular cells of more or less 9 by 7 km (in the centre of the domain) whereas EMEP uses a regular grid
of 0.1 by 0.1 degrees, corresponding to rectangular cells of more or less 7 by 11 km.
• Emissions: The CHIMERE emission reference year is 2010 with a gridding based on the EC4MACS project
proxies (Terrenoire et al., 2015) while EMEP uses a JRC set of emissions (Trombetti et al., 2017) based on
2014 as reference year.
• Boundary conditions: The CHIMERE domain extends from 10.5° East to 37.5° West and between 34° and
62° North while the EMEP domain extends from 30° East to 90° West and between 30° and 82° North.
• Meteorology: The two models use a different reference meteorological year; 2009 for CHIMERE and 2014
for EMEP; both meteorological fields are modelled through the Integrated Forecasting System (IFS) of
ECMWF.
• Model Parameterization: Apart from the vertical and/or horizontal resolutions, transport, deposition,
chemical processes are reproduced with different levels of complexity in the two models.
More details on the model simulations and settings can be found in Clappier et al., 2015 and Pisoni et al., 2019. Some
of the validation results for the two model configurations (CHIMERE and EMEP) are briefly presented in the
Supplementary Material, showing similar performances in terms of comparison against observations. For CHIMERE
the relation between predictions and observations at background stations is best characterised by a line through the
origin with slope of 1.05, indicating a slight under-prediction. The standard error is 5.7 μg/m$^3$ and uniform over the
range of concentrations. The R2 is 0.9. Concentrations at traffic and industrial stations are underestimated by roughly
10%. For EMEP the relation between predictions and observations is best characterised by a power low with exponent
0.66. The data show a relative standard error constant over the range of concentrations and equal to 30%.
Concentrations at traffic stations are under-predicted by 9% and over-predicted at industrial sites by 7%. It is important
to note that the use of different input and model set-up (as listed before) represents the usual practice when air quality
models are used, at the local scale, to assess the impact of air quality plans. This is why it is important here to analyse
how this choice influences the results and the subsequent design of an air quality plan; an issue that is often not tackled
in the literature. Finally, differences can arise from the SRR approximations themselves, even if validation against
CTM simulations show similar results for the 2 considered model set-up (see Supplementary Material).
Starting from these configurations, two set of SRRs are built for yearly average PM2.5 concentrations, based
respectively on CHIMERE and EMEP data.
Before looking at the source allocation results, in the next section a brief description of the SHERPA methodology is
proposed.
**3. SHERPA methodology**
Starting from the simulations performed with CHIMERE and EMEP, two sets of SHERPA SRR are built. Here we
briefly summarise how the SHERPA methodology works; we refer to Pisoni et al., 2019 for more details.
In the SHERPA approach, the PM concentration change in receptor cell "j" is computed as follows:

$$\Delta PM_j = \sum_{p}^{N_{prec}} \sum_{i}^{N_{grid}} a_{ij}^p \, \Delta E_i^p \qquad (1)$$

where N$_{grid}$ is the number of grid cells within the domain, N$_{prec}$ is the number of precursors, $\Delta E_i^p$ are the emission
changes, and $a_{ij}^p$ are the unknown parameters to be identified, representing the transfer coefficients between each
source cell i and receptor cell j. In SHERPA the $a_{ij}^p$ coefficients are cell-dependent, and assume a 'bell shape function'.
This bell shape function accounts for variation in terms of distance but is directionally isotropic, and can be defined
as follows:
$$a_{ij}^p = \alpha_j^p \left(1 + d_{ij}\right)^{-\omega_j^p} \quad\quad (2)$$
where $d_{ij}$ is the distance between a receptor cell "j" and a source cell "i". Thus, in SHERPA the matrix of transfer
coefficients is known when the two parameters $\alpha$ and $\omega$ are identified for a given receptor cell j and a given precursor
p (see Equation 2). The final formulation implemented in SHERPA is:

$$\Delta PM_j = \sum_p^{N_{prec}} \sum_i^{N_{grid}} \alpha_j^p \left(1 + d_{ij}\right)^{-\omega_j^p} \Delta E_i^p \quad\quad (3)$$

With the SHERPA approach, the key step is so to find the optimal $\alpha, \omega$ coefficients. As the number of unknown
parameters is equal to 2 $(\alpha, \omega)$ for each precursor and receptor cell "j", for the five precursors of PM2.5 (VOC –
volatile organic compounds, $SO_2$ – sulphur dioxide, $NO_x$ – nitrogen oxides, PPM – primary particulate matter and
$NH_3$ – ammonia), ten independent CTM simulations are needed for a given receptor cell. We refer to Pisoni et al.
(2018) and Thunis et al. (2016) for additional details about the SHERPA formulation and evaluation process.
Given its cell-to-cell characteristics (Equation 3), the SHERPA formulation can be used to assess the impact of
emission reductions over any given set of grid cells. Different geographical entities can therefore be freely defined in
terms of boundaries.
As mentioned earlier, the SHERPA approach is used in this work to analyse the differences in source allocation results
between two CTM: CHIMERE and EMEP, referred to in this paper as S-CHIMERE and S-EMEP, respectively. The
"S-" first letter in these acronyms reminds that we compare the EMEP and CHIMERE SRR rather than the models
themselves.

## 4.     Source allocation methodology

The aim of this work is to compare the main contributors to urban pollution in terms of sectors, geographical areas
and precursors, obtained with S-CHIMERE and S-EMEP. We focus on the PM2.5 yearly average concentrations as
target indicator, because PM2.5 is responsible for most of the health related burden in the EU urban areas (EEA 2019).
The approach is applied to 150 European cities, those analysed in the 'PM2.5 Urban Atlas' (Thunis et al., 2018).
As mentioned above, the cell-to-cell characteristics of the SHERPA approach allows assessing the impact of emission
reductions over any given set of grid cells (cities , regions or countries can be freely defined in terms of boundaries)
and emission reductions can be freely defined in terms of precursors or sectors. The following single (or combination
of) sectors, source areas and precursors are considered as sources.
In terms of sectors, the source categories follow the CORINAIR SNAP nomenclature for emissions:
162         •    Combustion in energy and transformation industries (SNAP 1),
163         •    Non-industrial combustion plants (SNAP 2),
164         •    Combustion in manufacturing industry (SNAP 3),

• Production processes (SNAP 4),
• Extraction and distribution of fossil fuels and geothermal energy (SNAP 5),
• Solvent use and other product use (SNAP 6),
• Road transport (SNAP 7),
• Other mobile sources and machinery (SNAP 8),
• Waste treatment and disposal (SNAP 9) and
• Agriculture (SNAP 10).
which have been aggregated in this work into five sectors:
• industry (SNAP 1, 3 and 4),
• residential (SNAP 2),
• traffic (SNAP 7),
• agriculture (SNAP 10), and
• others (SNAP 5, 6, 8 and 9).
In terms of geographical sources, four areas are considered for the analysis:
• the core city,
• the commuting zone,
• the rest of the country and
• international (what is outside the considered country).
The commuting zone is defined as the area surrounding the city where at least 15% of the population commutes daily
to the core city. The combination of the core city and the commuting zone is referred to as the functional urban area,
or FUA[2].
Finally, the precursors considered are $NO_X$, VOC, $NH_3$, PPM and $SO_2$.
This leads to 100 (4 areas x 5 precursors x 5 sectors) runs for each SRR and city. For small cities (66 out of 150) the
core city covers too few grid cells which would lead to discretization errors. In such case, the analysis is restricted to
the FUA. For these cities, 75 runs (3 areas x 5 precursors x 5 sectors) per city and model were therefore performed.
With 150 analysed cities for two CTM models, we note that the SHERPA approach allows for a comparison that
would have implied 26700 ((66x75 + 84x100) x 2 models) independent air quality simulations with a full CTM. The
same amount of runs with the SHERPA simplified model only takes few seconds per scenario. The results for S-
CHIMERE were published in the 'Urban PM2.5 Atlas' (Pisoni et al., 2018). In this paper, the same runs are done with
S-EMEP, and a comparison between the 2 is provided.
Each run performed with the SHERPA SRRs provides a concentration change ($\Delta C$) that results from an emission
reduction ($\Delta E$) with an intensity α applied to a given precursor, for a given sector and within a given area. The 'relative
potential' of a given precursor-sector-area source is expressed as $\Delta C/\alpha C$, (Thunis and Clappier, 2014). This indicator
represents the share of a particular emission source to the concentration. From a policy point of view, high 'relative
potential' sources are the ones to be addressed first to achieve the largest improvements. In this work, the SRRs $\Delta C$

---

[2]See https://www.oecd.org/cfe/regional-policy/functionalurbanareasbycountry.htm for details.

are obtained for emission reductions of $\alpha$=50%, a level that represents a threshold below which the quasi-linearity of
the model responses is preserved (Thunis et al., 2015). In other words, with this approach the model response in terms
of concentration change remains proportional to the emission change. It is important to stress that this threshold is
only valid for PM2.5 and for yearly average concentrations, as considered here. Because of this 50% threshold, it is
also worthwhile to note that the source allocation results discussed here provide information on the impact of potential
emission reductions up to that level, not beyond.

To compare the 'relative potentials' from S-CHIMERE and S-EMEP, we calculate the correlation. A high correlation
means that both models agree well on the emission sources (sectoral and/or geographic) that contribute most to the
concentration for a given city. The main advantage of a correlation indicator is that it ignores systematic differences.
In other words, if one model systematically predicts higher concentration changes for all sources than the other, this
is not detected by the correlation metric. This is a desirable characteristic because from a policy perspective, it is the
'relative ranking' among the sources contributions that counts rather than their absolute values.
**5.     Comparison of the results**
In this study, we compare the relative potentials for 150 cities, based on the two SHERPA implementations, S-
CHIMERE and S-EMEP. Source allocation is calculated at the city location characterised by the worst target indicator
value, i.e. the most polluted cell in the considered city. We first discuss the results for a few cities, before moving to
an EU wide perspective. Tables 1 to 4 show, for each emission area, sector and precursor, the 'relative potential'
expressed in percentage of the total concentration for the 2 models ('chimere_rp' and 'emep_rp') and the resulting
ranking in terms of importance ('emep.rank' and 'chimere.rank') for 4 cities: Liege, Genova, Turin and Madrid. These
cities are selected as representative samples to illustrate the characteristic behaviours obtained in our comparison. In
addition to this, Figures 1 to 4 show the S-CHIMERE/S-EMEP correlations obtained for various relative potentials
defined in terms of geographical area, sector, or their combinations. For Liege (Belgium), the overall (all individual
sectors, precursors and areas included, i.e. about 15000 relative potentials) Pearson correlation[3] between the relative
potentials of both SRR is the highest among the 150 cities (r=0.99, see Figure 1). Both models identify ammonia
emissions from agriculture, outside Belgium, as the main contributor to local PM2.5 concentrations. Primary PM from
local industry comes second and $NO_x$ from international traffic third. Although the lower ranked combinations are not
identical, they are quite similar. From a policy perspective, the fact that both SRR provide similar information is a
sign of robustness. It increases our confidence in the priority of interventions (which sectors-areas to act at first to
achieve the maximum air quality improvement). The values for the main sector-precursor-areas relative potentials are
reported in Table 1.

---

[3] The main aim of this work is to assess the policy implications (i.e. which source to tackle first) of using a model rather than another. This is why we focus on the ranking of the contributions (Pearson correlation) rather than on their absolute values.

**Table 1: Top 10 area-sector-precursor relative potentials to PM2.5 concentrations in Liege (B).**

| area | sector | precursor | emep_rp | emep.rank | chimere_rp | chimere.rank |
|---|---|---|---|---|---|---|
| International | Agriculture | NH3 | 22.9 | 1 | 20.6 | 1 |
| FUA | Industry | PPM | 12.6 | 2 | 12.4 | 2 |
| International | Road Transport | NOx | 7.5 | 3 | 6.9 | 3 |
| International | Industry | NOx | 4.9 | 5 | 5.2 | 4 |
| National | Agriculture | NH3 | 4.2 | 6 | 4.6 | 5 |
| International | Industry | SOx | 5.1 | 4 | 2.3 | 10 |
| International | Residential | PPM | 2.2 | 7 | 2.5 | 8 |
| FUA | Road Transport | PPM | 2.1 | 10 | 2.9 | 6 |
| International | Industry | PPM | 2.2 | 8 | 2.4 | 9 |
| FUA | Industry | SOx | 1.9 | 15 | 2.7 | 7 |
| International | Other | NOx | 2.2 | 9 | 1.9 | 13 |


A breakdown analysis for Liege is proposed in Figure 1 where correlations are calculated for relative potentials that
are aggregated in terms of sectors (5 relative potentials), area (4 relative potentials) or area/sectors (5 x 5 relative
potentials). In the case of Liege, all correlations are very good.

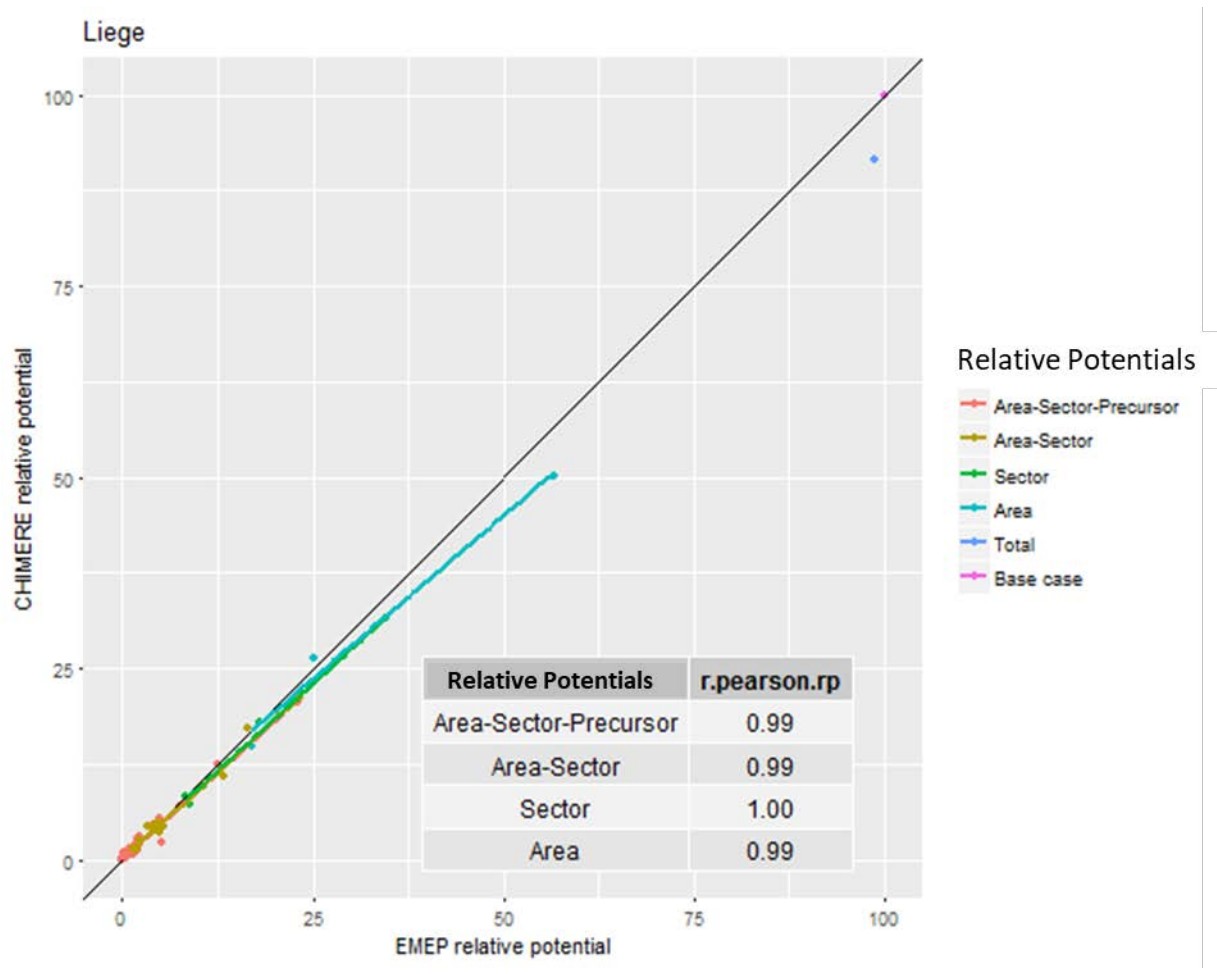

**Figure 1: Correlation between S-EMEP and S-CHIMERE relative potentials for different sector-area-precursor source aggregations in Liege (B).**

Unfortunately, the agreement is not always so good. For the city of Genova (Table 2 and Figure 2), both models agree that national/international ammonia emissions from agriculture are the largest contributor to local PM2.5 (see Table 2). But the third position in the priority ranking is occupied by $NO_x$ from national traffic for S-EMEP while it is PPM from the national residential sector for S-CHIMERE. However, the overall correlation still reaches 89% and the two main sources are similar. The agreement between the two models is therefore still satisfactory. It is interesting to note that for area-aggregated relative potentials, the correlation drops to 42%, highlighting possible differences in the way emission inventories are spatially distributed in the two models.

**Table 2: Top 10 area-sector-precursor relative potentials to PM2.5 concentrations in Genova (IT).**

| Relative Potentials | | | | | | |
|---|---|---|---|---|---|---|
| area | sector | precursor | emep_rp | emep.rank | chimere_rp | chimere.rank |
| National | Agriculture | NH3 | 14.5 | 1 | 11.3 | 1 |
| International | Agriculture | NH3 | 6.8 | 2 | 10.1 | 2 |
| National | Residential | PPM | 4.3 | 4 | 4.7 | 3 |
| FUA | Residential | PPM | 3.2 | 5 | 3.5 | 4 |
| National | Road Transport | NOx | 4.9 | 3 | 2.6 | 8 |
| FUA | Road Transport | NOx | 3.2 | 6 | 2.8 | 7 |
| International | Industry | SOx | 2.2 | 10 | 3.4 | 5 |
| National | Industry | SOx | 1.7 | 15 | 2.5 | 9 |
| International | Residential | PPM | 1.4 | 18 | 2.8 | 6 |
| FUA | Road Transport | PPM | 1.4 | 17 | 2.1 | 10 |
| FUA | Other | NOx | 2.5 | 8 | 0.7 | 21 |
| FUA | Industry | NOx | 2.4 | 9 | 0.0 | 59 |
| FUA | Industry | SOx | 3.1 | 7 | 0.0 | 62 |



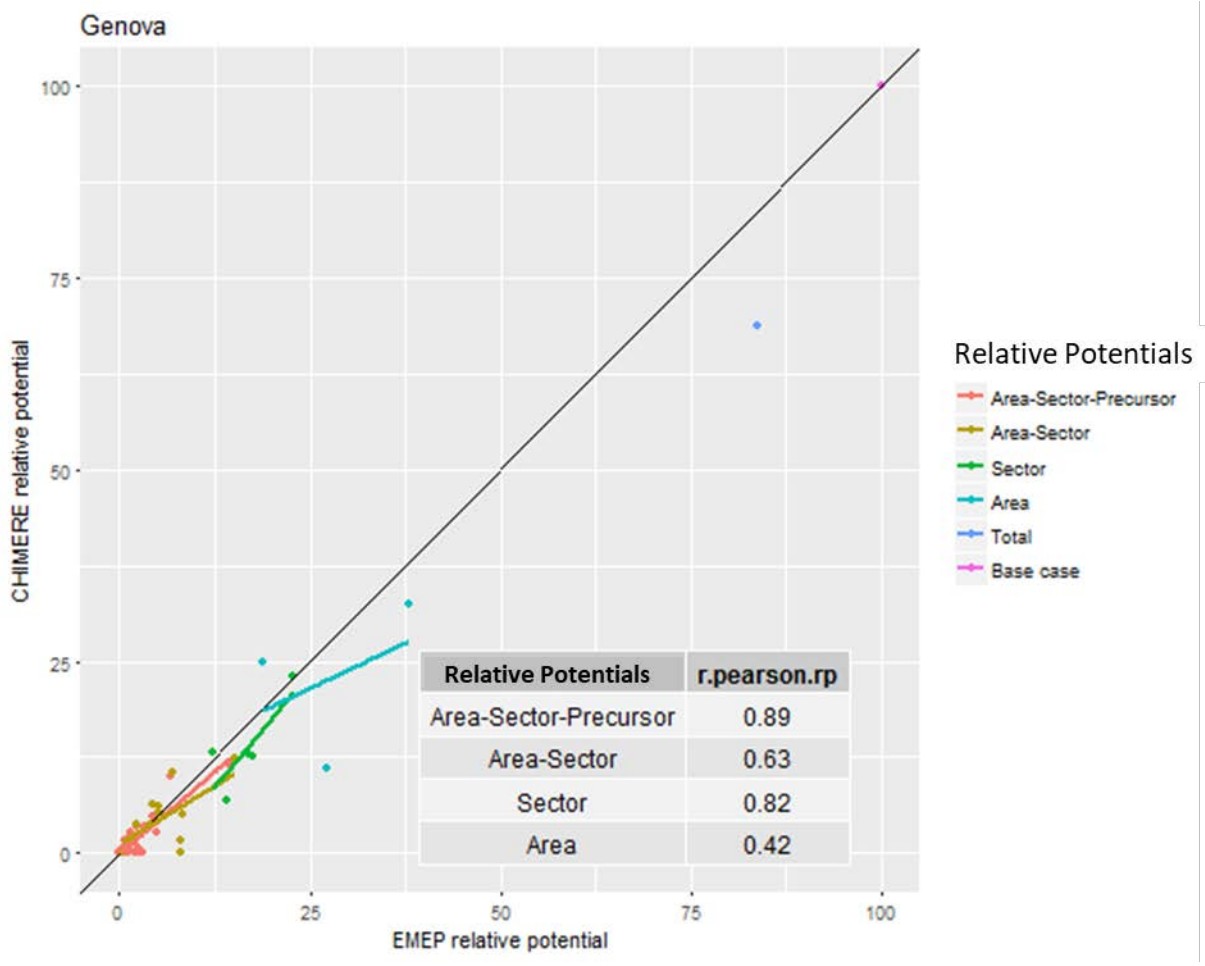


**Figure 2: Correlation between S-EMEP and S-CHIMERE relative potentials for different sector-area-precursor source aggregations in Genova (I).**


In the case of Torino (Table 3 and Figure 3), the two models give contradicting recommendations. While S-CHIMERE points to city residential heating as main contributor to PM2.5, S-EMEP points to national agriculture ammonia emissions. The model disagreement extends to the top 5 relative potentials. As indicated, the problem is probably related to the sectoral ($R^2$=0.78) rather than to the geographical dimension ($R^2$=0.97). Nevertheless, the overall correlation (0.81) is not too bad, and can be explained by the fact that the contribution values are not too different from each other (although the ranking is quite different).





**Table 3: Top 10 area-sector-precursor relative potentials to PM2.5 concentrations in Torino (I).**

| Relative Potentials | | | | | | |
|---|---|---|---|---|---|---|
| area | sector | precursor | emep_rp | emep.rank | chimere_rp | chimere.rank |
| FUA | Residential | PPM | 8.6 | 2 | 13.3 | 1 |
| National | Agriculture | NH3 | 10.6 | 1 | 5.9 | 4 |
| FUA | Industry | PPM | 6.4 | 3 | 13.3 | 2 |
| FUA | Road Transport | NOx | 6.2 | 4 | 4.8 | 6 |
| National | Residential | PPM | 4.9 | 7 | 5.4 | 5 |
| International | Agriculture | NH3 | 6.1 | 5 | 4.2 | 8 |
| FUA | Industry | NOx | 5.2 | 6 | 4.7 | 7 |
| FUA | Road Transport | PPM | 2.6 | 13 | 8.4 | 3 |
| FUA | Other | PPM | 2.9 | 12 | 3.5 | 10 |
| International | Residential | PPM | 2.0 | 16 | 4.0 | 9 |
| National | Road Transport | NOx | 4.3 | 8 | 1.3 | 18 |
| FUA | Residential | NOx | 3.8 | 9 | 1.0 | 23 |
| International | Road Transport | NOx | 3.1 | 10 | 0.8 | 25 |


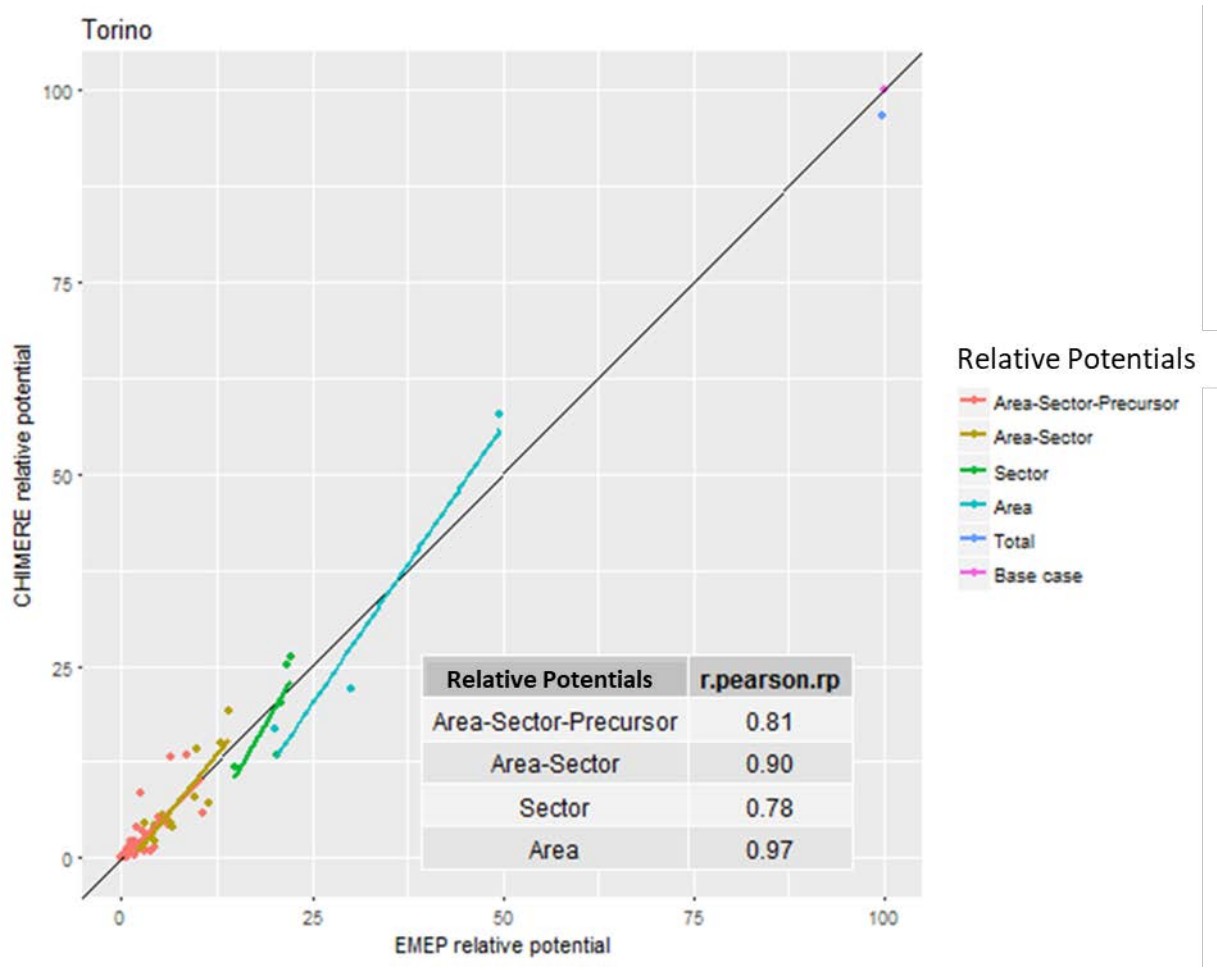

**Figure 3: Correlation between S-EMEP and S-CHIMERE relative potentials for different sector-area-precursor source**
**aggregations in Torino (I).**

In our last example (Madrid - Table 4 and Figure 4), differences are extremely important in terms of relative potentials
and ranking, leading to an overall correlation of 41%. All other correlations, with the exception of the spatial ones are
extremely poor. Uncertainties for this city are important, and the choice among policy options is not robust.

**Table 4: Top 10 area-sector-precursor relative potentials to PM2.5 concentrations in Madrid (E).**

| Relative Potentials | | | | | | |
|---|---|---|---|---|---|---|
| area | sector | precursor | emep_rp | emep.rank | chimere_rp | chimere.rank |
| City | Road Transport | PPM | 9.9 | 2 | 24.6 | 1 |
| City | Residential | PPM | 6.2 | 3 | 8.9 | 2 |
| City | Other | PPM | 2.0 | 9 | 5.0 | 4 |
| National | Agriculture | NH3 | 2.5 | 6 | 2.4 | 8 |
| Comm | Road Transport | PPM | 1.7 | 11 | 5.3 | 3 |
| National | Agriculture | PPM | 0.9 | 13 | 4.3 | 5 |
| City | Industry | PPM | 2.4 | 7 | 1.4 | 12 |
| City | Other | NH3 | 2.3 | 8 | 1.8 | 11 |
| Comm | Residential | PPM | 1.0 | 12 | 2.3 | 9 |
| City | Industry | SOx | 25.4 | 1 | 0.8 | 21 |
| City | Road Transport | NOx | 0.8 | 16 | 2.7 | 6 |
| City | Residential | SOx | 4.7 | 4 | 0.9 | 20 |
| National | Residential | PPM | 0.7 | 18 | 2.4 | 7 |
| National | Road Transport | PPM | 0.8 | 15 | 2.2 | 10 |
| National | Industry | SOx | 1.8 | 10 | 0.8 | 22 |
| Comm | Industry | SOx | 2.8 | 5 | 0.4 | 28 |



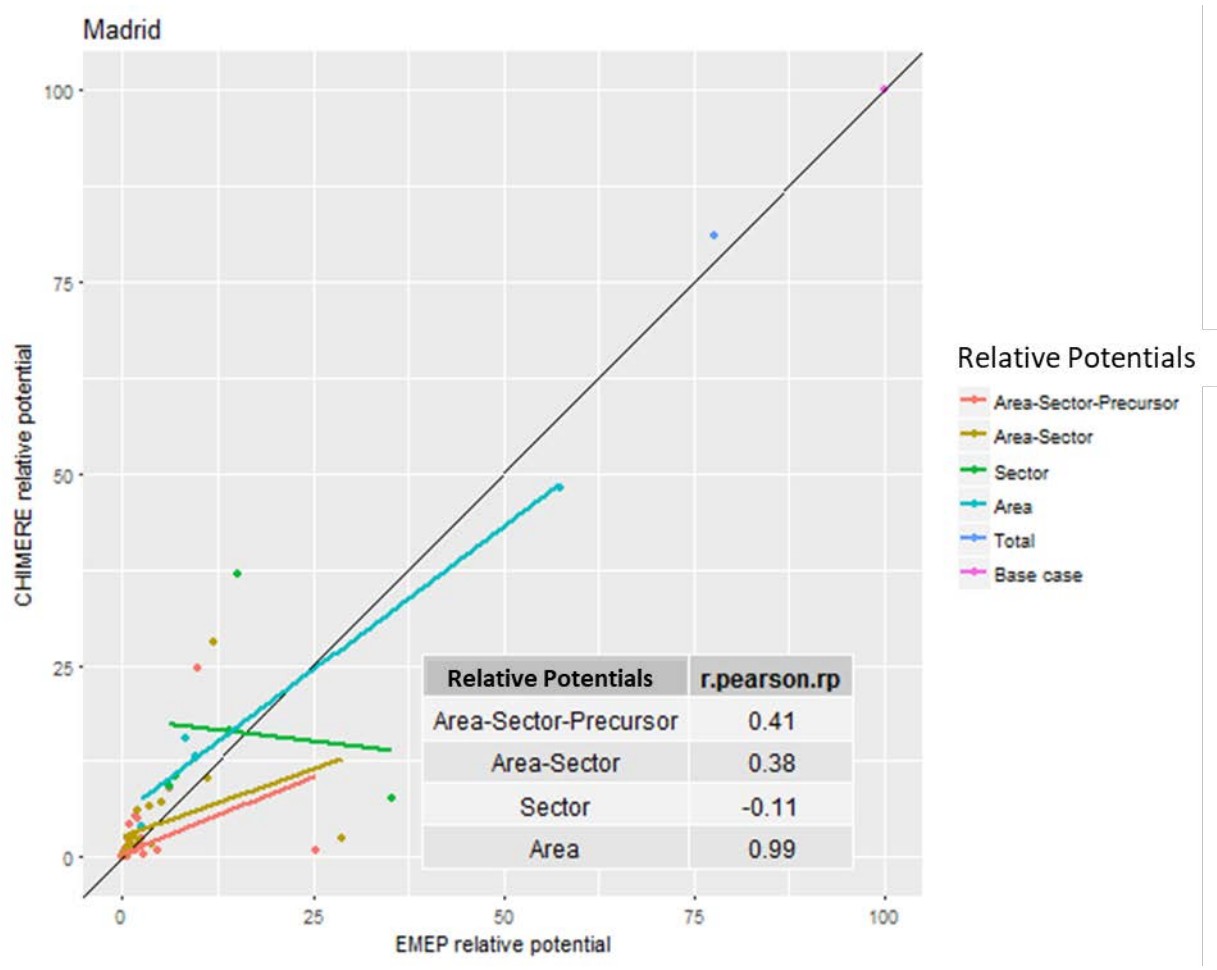

**Figure 4: Correlation between S-EMEP and S-CHIMERE relative potentials for different sector-area-precursor source**
**aggregations for Madrid (E).**

As seen from the city examples presented above, we can have both strong (Liege) and weak (Madrid) agreement
between the two modelling set-up.

The analysis presented above was done for all 150 cities, and we can here present the results in an aggregated way.
We will consider here that an overall correlation is very good above 95%, good between 90 and 95%, fair between 85
and 90%, poor between 70% and 85% and very poor below 70%. This is an arbitrary choice, but it is useful to start
grouping and classifying the results. The histogram of the overall correlations for all 150 cities (Figure 5:) shows that
the model agreement is good or very good for about half of the cities, satisfactory for another 21%, leaving 32% of
doubtful/problematic cities.

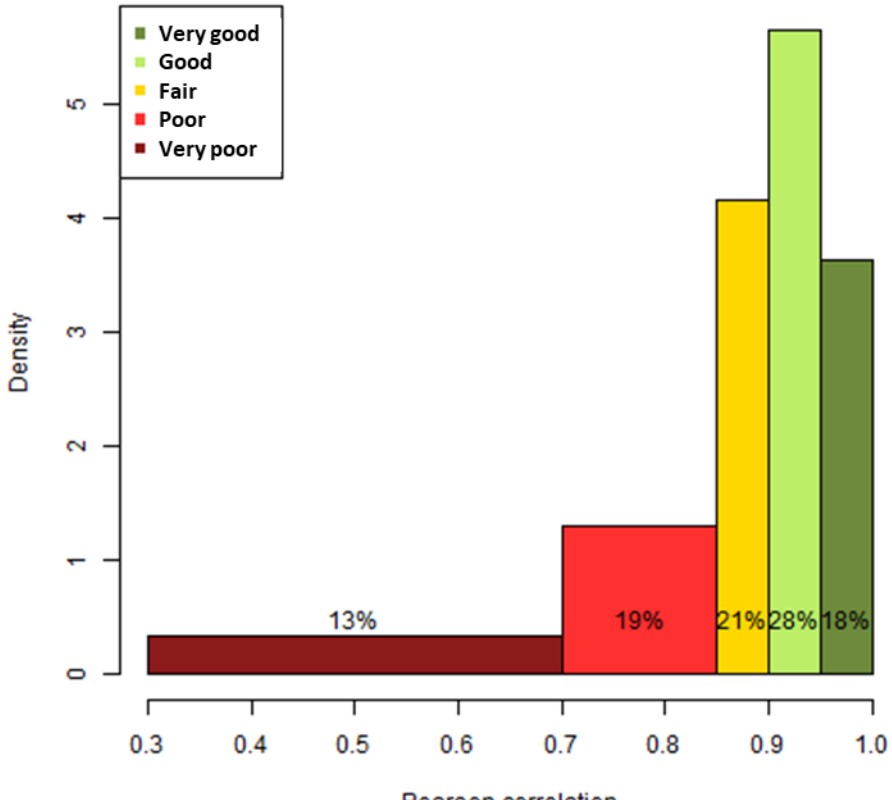

**Figure 5: Distribution of the Pearson correlation coefficients between relative potentials, for 150 cities.**

The mapping of the overall correlations (Figure 6) shows that cities with the highest variability are mostly located in Spain, Northern Italy and in the Baltic countries. For these areas, meteorological factors, emissions, and/or the impact of these input on concentrations in the air quality models, is larger than in other areas. In the Supplementary Material we show that even for the base case, results are quite different for countries like Spain. This might also have an impact on the correlation results shown in this Figure.

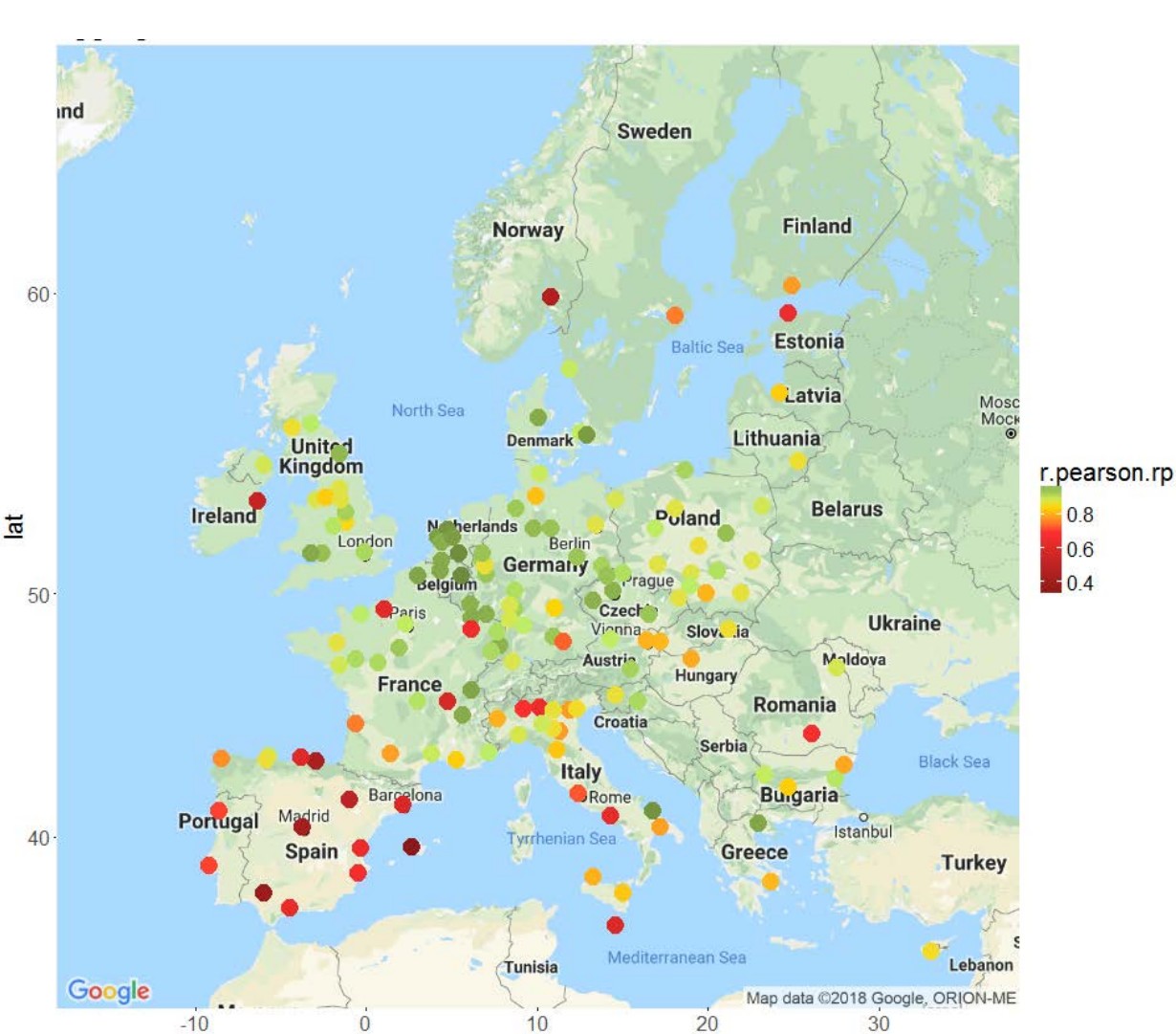

**Figure 6: Pearson overall correlation between EMEP and CHIMERE relative potentials.**

To the knowledge of the authors, this is one of the first attempts to systematically compare the sources and causes of pollution in European cities, using a harmonized approach. The reasons for the differences between cities highlighted above are however not easy to identify. This is because the SRRs used in this study are based on different meteorological years (2009 vs 2014), emissions (2010 vs 2014) and air quality models (CHIMERE vs EMEP). Although this analysis provides an overall estimate of the variability between policy responses and does not allow identifying the specific cause for the observed differences, it indicates where modelling improvements need to be

made. Modelling inconsistencies are indeed categorised in terms of geographical area, sectors and precursors, a useful
information to trigger discussion among modelling groups and direct the investigations towards the most problematic
issues.
It is also worth reminding that using different input and model set-up represents the usual practice whenever air quality
models are used at the local scale to assess the impact of air quality plans. Indeed, each local/regional authority
generally uses its own set of data and applies its own model. Therefore, only a single meteorology, a single emission
inventory for a single reference year and a specific model are used to identify the sources of pollution to target. The
impact of these choices on source allocation and on the subsequent design of an air quality plan is an issue that is not
often tackled.
It is probably unreasonable to think that a local authority can evaluate in a comprehensive way the variability of a
particular modelling pathway (too demanding in terms of sensitivity analysis). We however believe that this work can
be used to develop further guidance to select the proper modelling set-up (choice of meteorological year, emission,
model to use) to reduce the uncertainty attached to the results and increase their robustness.
The ultimate goal of this work would be to help decision makers to properly define key sources, so that only 'no-
regret' policies are selected. As mentioned above, the present approach flags out potential issues and a possible lack
of robustness (by quantifying the overall variability) but it cannot provide explanations for the observed differences.
The only process to identify the causes of differences, is to perform regular inter-comparison exercises where the
responses of models to emission changes are systematically tested via sensitivity analysis. While exercises of this type
occurred in the past years (Colette et al., 2017, Cuvelier et al., 2007, Pernigotti et al., 2013), it is crucial that these are
performed on a regular basis as models and input data continuously evolve.
**6.       Conclusions**
Before applying emission reduction measures to improve air quality, it is important to evaluate the importance of the
key sources contributing to pollution in a given area. The main methodology to perform this task is referred to as
'source allocation'.
Source allocation can be implemented in various ways. In this paper we use the SHERPA model, a source-receptor
relationship mimicking the behaviour of a fully-fledged CTM. With SHERPA one can perform hundreds of
simulations in few minutes to test the impact of various geographical, sectoral or precursor-based emission sources,
on the concentration at a location of interest. The result is a complete source-allocation study for a given domain
explaining the key sources of pollution at a given location.
In this work, we developed two SHERPA versions, based on two modelling set-up using different meteorological
reference year, emission inventories and air quality models. Even if these setting are quite different and difficult to
compare, they represent what happens in the real-world when designing air quality plans. Indeed, local authorities in
Europe are free to use different reference meteorological years, emissions and models. The comparison of these results
therefore provide an estimate of the variability attached to source allocation results for a given area.
The results can also be used to provide further guidance to define the modelling set-up and understand how this choice
impact the selection of priorities when designing air quality plans.

The two SHERPA SRRs versions (based on CHIMERE and EMEP) have then been used to perform source allocation
on 150 main cities in Europe, and results have been presented in terms of priorities of interventions (i.e.: which are
the sector/geographical areas/pollutants that are more relevant for air quality in a given city?).
The results are consistent for some cities, i.e. the modelling set-up produces the same ranking in terms of contributions,
whereas for other cities (about 30%) the two SRRs deliver different results. Even if it is not possible in this work to
identify the causes for these differences as additional sensitivity simulations would be needed for this, this work
indicates where modelling improvements need to be made. Modelling inconsistencies are indeed categorised in terms
of geographical area, sectors and precursors, a useful information to trigger discussion among modelling groups and
direct the investigations towards the most problematic issues. Although differences in terms of results were expected
(different assumptions deliver different results), it is comforting to see that similar policy decisions would be taken in
about 75% of cities considered in this study.
Thanks to the limited number of required simulations to build SHERPA, future work could envisage the
implementation of 'constrained setting' to build SRR (i.e. keeping the same air quality model but changing emissions,
or keeping the same emissions but changing the model) to be able to discriminate the role of these factors. Also, further
model inter-comparison works should be fostered.
**Code and data availability**
The code and data used to perform the analysis presented in this paper is available in a Zenodo repository (Degraeuwe
et al., 2020). Also the SHERPA model, providing the source-receptor relationships applied in this paper, is available
in another Zenodo repository (Degraeuwe et al., 2020b).
**Authors contribution**
BD developed the methodology, performed the analysis and drafted a first version of the paper. PT conceived the
initial development of SHERPA, and contributed to the structuring and revision of the paper. EP developed the
SHERPA tool, contributed to the interpretation of the results and to the preparation of the final version of the paper.
**Acknowledgements**
We acknowledge A. Colette (INERIS), H. Fagerli and S. Tsyro (The Norwegian Meteorological Institute) for their
work in performing CTM simulations, and for exchange of views on the content of this paper.

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
