# Peer review of "Prioritising the sources of pollution in European cities: do air quality modelling applications provide consistent responses?"

_Geoscientific Model Development, 2020_

## Referee Comment (RC1) · Anonymous Referee #1 · 24 Jun 2020

The paper address relevant scientific modelling questions regarding the application and feasibility o fair quality models to support air quality plans, which is quite new and advanced research. The methodology is appropriate and well-described. Nevertheless, there are some major (and minor) critical points that should be addressed before publication. They are listed below.

Major changes Line 70-77: please comment about the different baseline year for the emission and meteorology data and its implication on the results. Authors could highlight here (what is said at the end of the paper) that these differences in input data are interesting to the analysis of results since this is the usual way to define air quality plans, etc... Nevertheless, these differences should be analysed in detail in order to understand their role in the SHERPA results. At least, authors should give some information regarding the validation (with observational data) of the 2 different models application (EMEP and CHIMERE). It is different if we are talking about 2 models with good performance or 2 models with completely different skills... Line 249: Again, regarding the sentence "Probably for these areas the differences in terms of meteorology, emissions, and their impact on concentrations through the air quality models, is higher than in other areas." This should be explored and analysed to better support the interpretations and conclusions and shouldn't be only an hypothesis to mention...

Minor changes Abstract: please add more details in the last sentence ("But there are also cases where results are contradictory". it is not mention which was the pollutant studied: PM2.5 Line 18: all instead of al Line 19: Please review the sentence: "FAIRMODE (the Forum for air quality modelling in Europe) i.e. provides tools to assess the..." Line 82: please explain why PM2.5 is the focus, and why only this one Line 145: please use subscript on the compound's chemical formulas Line 182: "chimere.rank" Line 184: what do the author mean with "for the different types of considered aggregations (area, sector, area-sector, ...)"? It is not obvious Line 185: Before starting to analyse the results for specific cities, the authors should identify and present which were the 4 cities selected (and their different behaviours associated) Line 194/209/...: Tables caption should be on the top of the table

---

## Referee Comment (RC2) · Anonymous Referee #2 · 30 Jun 2020

Degraeuwe et al. describe the application of the SHERPA technique for determining Source/Receptor Relationships (SRRs) to the assessment of mitigation options for annual average PM2.5 concentrations in 150 European cities. SRRs are calculated from the output of two Chemical Transport Models (CTMs), CHIMERE and EMEP, which are commonly used in Europe for air quality simulation. The benefit of using pre-calculated SRRs instead of directly using the CTMs themselves is that the SRRs effectively emulate the relationship between emissions in each CTM grid cell and concentrations in other grid cells without having to simulate the full set of physical and chemical processes involved. SHERPA in particular provides an efficient way of calculating cell-to-cell SRRs without having to run a large number of training simulations, by making some

assumptions about the degree to which grid cells can influence each other based on their separation.

The authors use the two different sets of SRRs to determine the most effective options for mitigation of annual average PM2.5 in the 150 selected cities. They find that despite the use of different CTMs, emission inventories, and base meteorological years, the mitigation options identified for each of the cities are generally very similar. A few cases are however identified where the use of the different SRRs produces contradictory recommendations.

While the topic is certainly within the scope of GMD, and the results as presented should be of interest to the community, it seems to me that the authors have gone to an extremely minimal amount of effort with this manuscript. The quality of the manuscript in its present form is not high enough to meet the standards that this reviewer would expect from GMD. Major revisions are required before the manuscript can be published.

Firstly, the authors appear to cite mostly their own work, or the work of their colleagues. This approach may be acceptable for an internal technical report, but in the peer-reviewed literature, authors must place their work in the broader context of the work that has come earlier, and clearly explain its novelty. The use of SRRs in air quality assessment has been prevalent for a long time, and SHERPA is not the only way that exists to calculate SRRs. It is not the job of this reviewer to perform the literature survey that the authors of this manuscript have neglected, so I will not suggest any specific references. But more context is certainly needed, and not only in the introduction; while the results are new and interesting, this is no excuse for not discussing them with appropriate reference to the existing literature.

Secondly, for a technical journal such as GMD, the paper is extremely short on technical detail. In Section 3, the reader is referred to Pisoni et al. (2019) for all but a few of the relevant details. Of course the reference is appropriate in this section, but the paper should also contain enough detail to stand on its own. The authors need to

summarise the key points from this earlier work. For example, readers need to know how the SHERPA technique differs from other approaches to calculating SRRs, and how well it has been shown to work. Have mitigation options identified with SHERPA been compared with actual CTM simulations of the same mitigation options? What are the strengths and weaknesses of the approach as identified by earlier work, and what are their implications for the present manuscript?

I also have a couple of minor comments.

It would be nice to see a short explanation of how the four cities shown in detail were chosen. It's good to see an example of a situation in which the approach works well, and a situation in which it doesn't (Liege and Madrid). But what about the other two cities (Genova and Torino)? Were these chosen to highlight specific points? Or for some other reason?

For the cases when the use of the two sets of SRRs from different CTMs yields different mitigation options, the authors take the position that their method is simply unable to explain the differences. I find this somewhat lazy. Actually the disagreement could point the way to targeted CTM simulations (or other analysis) designed to specifically understand the relevant processes. It would add a lot to the paper to see some more discussion of this.

---

## Author Comment (AC1) · 6 Jul 2020

We thank the reviewer for the comments.

We address in our reply all the comments, and in particular the one related to the differences between the 2 considered model set-up, with validation against observations.

In particular, please find attached: - a 'reply to reviewer's comments' document - a 'supplementary material' document, with base case validation results for the 2 model set-up - a html, with further analysis performed on the data, and used to further explore differences in input data for the 2 model set-up

[Figure]

Please also note the supplement to this comment:
https://gmd.copernicus.org/preprints/gmd-2020-90/gmd-2020-90-AC1-supplement.zip

—————————————————————

Interactive
comment

---

## Short Comment (SC1) · 7 Jul 2020

This paper presents a comparison between the results obtained with two different set up of the SHERPA Source Receptor Relationship (SRR): S-CHIMERE and S-EMEP. Each of these two SHERPA configurations is used to compute the impact of different emission reductions (per activity sectors, per areas and per precursors) for 150 cities in Europe. The authors compare all the impacts provided by the two SHERPA configurations to evaluate the variability resulting from the use of two model systems (CHIMERE and EMEP). This work is without any doubts very interesting because it provides information about the robustness of model results which could be directly

used by decision makers to design abatement strategies. The authors take advantage of the capacity of SHERPA to simulate a very large number of scenarios concerning spatial as well as sectorial emission reductions. 150 cities have been considered and 100 scenarios have been computed for each of these cities. As far as I know, SHERPA is the only tool able of such performances and it is the first time that so many cities and scenarios have been tested. This is why I think that the most interesting results of this article concerns the analysis of all cities and all scenarios (graphic of figure 5 and map of figure 6). The graphic of figure 5 and the map of figure 6 shows that a large part of the impacts computed by the two SHERPA configurations are closed to each other. 67% of the 150 cities are evaluated as Fair, Good or Very Good (Pearson coefficients above 0.85 in figure 5). Moreover, these cities are located in the largest part of Europe (all Europe except the Iberian Peninsula, southern Italy, extreme North Europe and some points like Milan or Lyon). It indicates that the results are robust, which may reassure decision-makers. Unfortunately, even if two models give similar results, they can both be wrong. For this reason, a diagnosis of good robustness remains difficult to exploit. On the contrary, large differences between the results of two models shows that, at least, one of the models is wrong. In such case, the information provided by the comparison may worry decision-makers but become very valuable for model developers and data providers. Observing the map of figure 6 shows clearly that the Iberian Peninsula and the southern Italy are not well simulated by at least one of the SHERPA configurations. This should encourage the developers of CHIMERE and EMEP to control their models and their data in these regions. I advise the authors to insist on this point which seems to me one of the major contributions of their work. But the evaluation of the difference between two CTM like EMEP and CHIMERE required some wariness. Indeed, SHERPA does not reproduce exactly the results of a CTM generating some errors which will be probably different for EMEP and CHIMERE. The differences which appear between EMEP and CHIMERE will be amplified or damped by SHERPA. So that, high differences between the two SHERPA configurations could hide low differences between EMEP and CHIMERE and vice et versa. This problem

has not been commented and is even not mentioned in this article. I advise the authors to address this point. I suppose they can easily refer to the SHERPA accuracy that have been estimated in their previous publications. The authors use the Pearson correlation to evaluate the differences between the two SHERPA configurations which is perhaps not the best statistical indicator. The Pearson coefficient does not spot situations where the results of one of the models are proportional to the other. Let suppose, for example, that the results of one of the models is constantly twice the results of the other model. The Pearson coefficient will then be equal to 1. I advise the author to use another indicator, like the RMSE, it will probably not change their conclusions but should avoid the problem just mentioned. Then, it could be interesting to evaluate (even roughly) a threshold above which the differences observed between the two SHERPA configurations reflect significant differences between the two systems of models EMEP and CHIMERE. This would help locate the areas where the differences between EMEP and CHIMERE are proven with near certainty.

―――――――――――――――――

---

## Author Comment (AC2) · 9 Jul 2020

Dear reviewer

please find attached - a reply to the reviewer's comments - a new supplementary material, with further details on the two source-receptor implementations, as requested - an html file, containing further analysis on basecase concentrations for the 2 reference years considered in the manuscript

Thanks again for your comments Kind regards Enrico

Please also note the supplement to this comment:
https://gmd.copernicus.org/preprints/gmd-2020-90/gmd-2020-90-AC2-supplement.zip
* * *

---

## Author Comment (AC4) · 27 Jul 2020

reply (see response done on the 10th of July to the same reviewer, thanks)

---

## Editor Comment (EC1) · Fiona O'Connor (Editor) · 30 Sep 2020

Please also note that the manuscript, as it currently stands, does not fully comply with the requirements of GMD. In particular,

1. Analysis code is on GitHub, which is not a persistent archive. GitHub themselves tell authors to use Zenodo for this: https://guides.github.com/activities/citable-code/

2. No publicly archived source code for Sherpa. We just get a link to their website where it appears one can register to download a binary. The source code of the exact version used needs to be publicly and persistently archived. For more details the

authors should see the policy below.

3. The code and data citations are plain urls in the text. This is poor practice. Code and data should appear as full references in the bibliography with citations in the text. Many good archives (including Zenodo) will help with this by providing the correct BibTeX and other reference manager entries on the archive page itself.

Further details on code and data availability requirements are in the GMD model code and data policy: https://www.geoscientific-model-development.net/about/code_and_data_policy.html. The reasons for the policy and more detail are provided in this editorial: https://doi.org/10.5194/gmd-12-2215-2019.

---

## Author Comment (AC5) · 1 Oct 2020

Dear Editor

I tried to fix the issue you mention in your comment online, uploading the codes to Zenodo and modifying the references to the code in the manuscript...

in particular adding these 2 codes / citations:

- Degraeuwe B., Pisoni E., & Thunis P.: Routines and data to compare different source-receptor relationships results. (Version v1.1). http://doi.org/10.5281/zenodo.4059786, 2020.

- Degraeuwe B., Pisoni E., & Thunis P.: Source code for the SHERPA source receptor relationships. (Version v1.0). http://doi.org/10.5281/zenodo.4059770, 2020b.

let me know if this solves the issue

Bests Enrico

---

## Author Response (AR1)

**Prioritising the sources of pollution in European cities: do air quality modelling applications provide consistent responses?**

- 3 Bart Degraeuwe, Enrico Pisoni, Philippe Thunis
- 4 European Commission, Joint Research Centre (JRC), Ispra, Italy
- 5 *Correspondence to*: E. Pisoni (enrico.pisoni@ec.europa.eu)

**6 Anonymous Referee #1**

Received and published: 24 June 2020

9 The paper address relevant scientific modelling questions regarding the application and feasibility 10 of air quality models to support air quality plans, which is quite new and advanced research. The 11 methodology is appropriate and well-described. Nevertheless, there are some major (and minor) 12 oritigal points that about the addressed before publication. They are listed below.

critical points that should be addressed before publication. They are listed below.

Major changes Line 70-77: please comment about the different baseline year for the emission and meteorology data and its implication on the results. Authors could highlight here (what is said at the end of the paper) that these differences in input data are interesting to the analysis of results since this is the usual way to define air quality plans, etc:...Nevertheless, these differences should be analysed in detail in order to understand their role in the SHERPA results. At least, authors should give some information regarding the validation (with observational data) of the 2 different models application (EMEP and CHIMERE). It is different if we are talking about 2 models with good performance or 2 models with completely different skills

We thank the reviewer for the comments. As suggested, we highlighted (at the beginning of

Section 2) the importance of accounting for the variability of the input data (in terms of emissions,
 meteorology, etc...) when assessing potential impacts, something that is not performed in practice for air quality plans that are often based on a unique set of input. Also, as suggested by the reviewer, we added graphs in the Supplementary Material with model base-case validations

- (against observations) for the CHIMERE and EMEP configurations, that show similar skills. We
   however highlight here the fact that similar behaviour on base case concentrations do not imply
- 29 similar source contributions (see Supplementary Material).
- 30 In section 2, we propose to add this text:

"Validation results for the two model configurations are presented in the Supplementary Material,
 showing similar performance (for CHIMERE and EMEP) in terms of comparison against observations. For CHIMERE the relation between predictions and observations at background

- 34 stations is best characterised by a line through the origin with slope of 1.05, indicating a slight
- under-prediction. The standard error is 5.7 μg/m3 and uniform over the range of concentrations.
- 36 The R2 is 0.9. Concentrations at traffic and industrial stations are underestimated by roughly 10%.
- 37 For EMEP the relation between predictions and observations is best characterised by a power
- 38 low with exponent 0.66. The data show a relative standard error is constant over the range of concentrations and equal to 30%. Traffic stations are under-predicted by 9% and industrial 40 stations over-predicted by 7%.

- 41 It is important to note that the use of different input and model set-up (as listed before) represents
- 42 the usual practice when air quality models are used, at the local scale, to assess the impact of air
- 43 quality plans. This is why it is important (in this manuscript) to analyse how this choice influences
- 44 the results and the subsequent design of an air quality plan; an issue that is often not tackled in the scientific literature. Some differences in results might be due to trends in emissions and 46 concentrations between 2010 and 2014. During this period, concentrations at Airbase stations 47 decreased 2.2% per year on average ( $\sigma$  = 2.7%/year). Starting from these configurations, two set 48 of SRRs have been built to model yearly average PM2.5 concentrations, based respectively on

CHIMERE and EMEP data. The focus of this study is on PM2.5 yearly averages, as this is the pollutant with the highest impact on human health, and a key focus of policy makers in Europe.

Before looking at the source allocation results, in the next section a brief description of the 52 SHERPA methodology is proposed."

Line 249: Again, regarding the sentence "Probably for these areas the differences in terms of meteorology, emissions, and their impact on concentrations through the air quality models, is higher than in other areas." This should be explored and analysed to better support the interpretations and conclusions and shouldn't be only a hypothesis to mention.

We elaborated a bit more in this section, linking also to the validation of the base case for 2 models 59 setup, now presented in the Supplementary Material.

We propose to modify the paper as follows:

"Probably for these areas the differences in terms of meteorology, emissions, and their impact on concentrations through the air quality models, is higher than in other areas (in the Supplementary

Material we show i.e. how the validation results, for the base case, are quite different for Spain in the 2 model implementation, and this could also have an impact on the correlation results shownin Figure)."

Minor changes Abstract: please add more details in the last sentence ("But there are also cases 68 where results are contradictory". it is not mention which was the pollutant studied: PM2.5

- 69 We clarify now, in the abstract, that the paper is on PM2.5 yearly averages.
- 70 The paper has been modified as follows:
- "But there are also cases where results (in terms of source allocation for PM2.5 yearly averages)
   are contradictory."
- 72 are contradictor
- 74 Line 18: all instead of al
- 75 We fixed the typo.
- 76

Line 19: Please review the sentence: "FAIRMODE (the Forum for air quality modelling in Europe)i.e. provides tools to assess the:"

The sentence has been reviewed, and modified as follows:

*"For example, FAIRMODE (the Forum for air quality modelling in Europe) provides tools to assess*

- 81 the quality of the models, as the Model Quality Indicator and Model Quality Objective (Pernigotti
- 82 el al., 2013b; Viaene et al., 2016)."

**Line 82: please explain why PM2.5 is the focus, and why only this one**

We specified that the focus is PM2.5, because we want to concentrate on the pollutants with the highest burden on human health. We also stress the fact that because a large number of sources

- 87 contribute to PM2.5 concentrations, this is the most challenging pollutant to manage in air quality
- plans. It is therefore important to assess the different model contributions for that pollutant inparticular.
- 90 This is how we propose to modify the text:
- 91 "The focus of this study is on PM2.5 yearly averages, as this is the pollutant with the highest
- 92 impact on human health, and a key focus of policy makers in Europe. We also stress the fact that
- 93 because a large number of sources contribute to PM2.5 concentrations, this is the most
- 94 challenging pollutant to manage in air quality plans."
- 95

- 96 Line 145: please use subscript on the compound's chemical formulas
- 97 This issue has been fixed.
- 98

Line 182: "chimere.rank"

This issue has been fixed.101

- Line 184: what do the author mean with "for the different types of considered aggregations
- 103 (area, sector, area-sector, ...)"? It is not obvious
- 104 This has been now better explained in the text. Text has been modified as follows:
- 105 "In addition to this, Figures 1 to 4 show the 'relative potentials' for the 2 models (S-CHIMERE and
- 106 S-EMEP), for the different types of scenarios (considering emission reductions for the selected 107 geographical area, for the chosen sector, or for combinations of geographical areas - sectors, ...)
- 107 geographical area, for the chosen sector, or for combinations of ge
and their corresponding correlations, for the same cities."
- 108
- Line 185: Before starting to analyse the results for specific cities, the authors should identify and present which were the 4 cities selected (and their different behaviours associated)
- 112 Text has been rephrased to reflect the reviewer's comment. This has been explained in the text:
- 113 *"We present results for 4 cities (Liege, Genova, Turin and Madrid) that are representative of the different behaviours found in our results."*
- 115
- Line 194/209/: Tables caption should be on the top of the table
- 117 This issue has been fixed.
- 118
- 119

**120 Anonymous Referee #2**

- 121 Received and published: 30 June 2020
- 122

Degraeuwe et al. describe the application of the SHERPA technique for determining Source/Receptor Relationships (SRRs) to the assessment of mitigation options for annual 124 average PM2.5 concentrations in 150 European cities. SRRs are calculated from the output of 125 126 two Chemical Transport Models (CTMs), CHIMERE and EMEP, which are commonly used in Europe for air quality simulation. The benefit of using pre-calculated SRRs instead of directly 127 128 using the CTMs themselves is that the SRRs effectively emulate the relationship between 129 emissions in each CTM grid cell and concentrations in other grid cells without having to simulate 130 the full set of physical and chemical processes involved. SHERPA in particular provides an efficient way of calculating cell-to-cell SRRs without having to run a large number of training 131 simulations, by making some assumptions about the degree to which grid cells can influence each 132 other based on their separation. The authors use the two different sets of SRRs to determine the 133 most effective options for mitigation of annual average PM2.5 in the 150 selected cities. They find 134 that despite the use of different CTMs, emission inventories, and base meteorological years, the 135 mitigation options identified for each of the cities are generally very similar. A few cases are 136 137 however identified where the use of the different SRRs produces contradictory recommendations. 138 While the topic is certainly within the scope of GMD, and the results as presented should be of interest to the community, it seems to me that the authors have gone to an extremely minimal 139 140 amount of effort with this manuscript. The quality of the manuscript in its present form is not high 141 enough to meet the standards that this reviewer would expect from GMD. Major revisions are 142 required before the manuscript can be published.

Firstly, the authors appear to cite mostly their own work, or the work of their colleagues. This 144 approach may be acceptable for an internal technical report, but in the peer-reviewed literature, 145 146 authors must place their work in the broader context of the work that has come earlier, and clearly 147 explain its novelty. The use of SRRs in air quality assessment has been prevalent for a long time, and SHERPA is not the only way that exists to calculate SRRs. It is not the job of this reviewer to 148 149 perform the literature survey that the authors of this manuscript have neglected, so I will not 150 suggest any specific references. But more context is certainly needed, and not only in the introduction; while the results are new and interesting, this is no excuse for not discussing them 151 152 with appropriate reference to the existing literature. 153 As suggested by the reviewer, we extended the literature review. Note however that although many SRRs have been developed for air quality, we are not aware of a methodology that is flexible and fast enough to assess so many sensitivities at the urban scale. We stressed these points in the Introduction by adding the text below (for the discussion of the results and technical aspect, see our 'reply' to your next comment):

[revised manuscript text omitted]

- 201 approximate two CTMs: CHIMERE and EMEP and compare their responses."
- 202

Secondly, for a technical journal such as GMD, the paper is extremely short on technical detail. In Section 3, the reader is referred to Pisoni et al. (2019) for all but a few of the relevant details. Of course the reference is appropriate in this section, but the paper should also contain enough detail to stand on its own. The authors need to summarise the key points from this earlier work. For example, readers need to know how the SHERPA technique differs from other approaches to calculating SRRs, and how well it has been shown to work. Have mitigation options identified

- with SHERPA been compared with actual CTM simulations of the same mitigation options? What are the strengths and weaknesses of the approach as identified by earlier work, and what are
- 211 their implications for the present manuscript?
- As suggested by the reviewer, we now provide more details on the methodology and validation results of the SRR. In particular, in the Supplementary Material, we added information on the base case validation for the 2 model set-up (validation against observations), and also on how the
- 214 Case validation for the 2 model set-up (validation against observations), and also on now the 215 SRRs behave in comparison to CTM simulations (validation against CTM results). Please find
- attached to this reply, the Supplementary Material, with the aforementioned contents.
- Furthermore, in the manuscript we propose to add this text (in Section 2), to better detail thetechnical capabilities of the SRR, and validation results:
- "More details on the model simulations and settings can be found in Clappier et al., 2015 and
   Pisoni et al., 2019. Validation results for the two model configurations are presented in the
- 221 Supplementary Material, showing similar performances (for CHIMERE and EMEP) in terms of comparison against observations. For CHIMERE the relation between predictions and 222 observations at background stations is best characterised by a line through the origin with slope 223 of 1.05, indicating a slight under-prediction. The standard error is 5.7 µg/m3 and uniform over the 224 range of concentrations. The R2 is 0.9. Concentrations at traffic and industrial stations are 225 underestimated by roughly 10%. For EMEP the relation between predictions and observations is 226 best characterised by a power low with exponent 0.66. The data show a relative standard error 227 228 constant over the range of concentrations and equal to 30%. Concentrations at traffic stations are 229 under-predicted by 9% and over-predicted at industrial stations by 7%. It is important to note that the use of different input and model set-up (as listed before) represents the usual practice when 230 air quality models are used, at the local scale, to assess the impact of air quality plans. This is 231 232 why it is important (in this manuscript) to analyse how this choice influences the results and the 233 subsequent design of an air quality plan; an issue that is often not tackled in the scientific 234 literature. Some differences in results might be due to trends in emissions and concentrations between 2010 and 2014. During this period, concentrations in Airbase stations decrease yearly 235 236 by 2.2% on average ( $\sigma$  = 2.7%/year). Finally, differences can arise from the SRR approximation, 237 even if (as shown in the Supplementary Material) validation against CTM simulations show similar results for the 2 considered model set-up. Starting from these configurations, two set of SRRs 238 have been built to model yearly average PM2.5 concentrations, based respectively on CHIMERE 239 and EMEP data." 240

242

I also have a couple of minor comments. It would be nice to see a short explanation of how the four cities shown in detail were chosen. It's good to see an example of a situation in which the approach works well, and a situation in which it doesn't (Liege and Madrid). But what about the other two cities (Genova and Torino)? Were these chosen to highlight specific points? Or for some other reason?

In section 5, we propose to add this text:

"Figures 1 to 4 show the 'relative potentials' for the 2 models (S-CHIMERE and S-EMEP), for the
different types of performed scenarios (considering emission reductions for the selected
geographical area, for the chosen sector, or for combinations of geographical areas - sectors, ...)
and their corresponding correlations, for the same cities. We present results for 4 cities (Liege,
Genova, Turin and Madrid) that are representative of the different behaviours found in our results."

For the cases when the use of the two sets of SRRs from different CTMs yields different mitigation options, the authors take the position that their method is simply unable to explain the differences. I find this somewhat lazy. Actually the disagreement could point the way to targeted CTM simulations (or other analysis) designed to specifically understand the relevant processes. It would add a lot to the paper to see some more discussion of this.

We now better explain the possible reasons for disagreement, referring to the Supplementary
 Material. Even if further investigation would be required to understand precisely why these
 differences occur.

We propose to add this text, at the end of Section 5:

*"The overall correlation map of Europe (Figure 6) shows that cities with the highest variability are mostly located in Spain, Northern Italy as well as the Baltic countries. For these areas,*

- 267 meteorological factors, emissions, and/or the impact of these input on concentrations in the air
- 268 quality models is higher than in other areas. In the Supplementary Material we show i.e. how the
- 269 validation results, for the base case, are quite different for Spain in the 2 model implementation,
- 270 and this could also have an impact on the correlation results shown in the Figure."
- 271 272

**273 Referee #3**

- 274 Received and published: 7 July 2020
- 275

This paper presents a comparison between the results obtained with two different setup of the 277 SHERPA Source Receptor Relationship (SRR): S-CHIMERE and S-EMEP. Each of these two 278 SHERPA configurations is used to compute the impact of different emission reductions (per 279 activity sectors, per areas and per precursors) for 150 cities in Europe. The authors compare all the impacts provided by the two SHERPA configurations to evaluate the variability resulting from 280 281 the use of two model systems (CHIMERE and EMEP). This work is without any doubts very 282 interesting because it provides information about the robustness of model results which could be 283 directly used by decision makers to design abatement strategies. The authors take advantage of 284 the capacity of SHERPA to simulate a very large number of scenarios concerning spatial as well as sectorial emission reductions. 150 cities have been considered and 100 scenarios have been 285 286 computed for each of these cities. As far as I know, SHERPA is the only tool able of such performances and it is the first time that so many cities and scenarios have been tested. This is 287 288 why I think that the most interesting results of this article concerns the analysis of all cities and all scenarios (graphic of figure 5 and map of figure 6). The graphic of figure 5 and the map of figure 289 6 shows that a large part of the impacts computed by the two SHERPA configurations are closed 290 to each other. 67% of the 150 cities are evaluated as Fair, Good or Very Good (Pearson 291 292 coefficients above 0.85 in figure 5). Moreover, these cities are located in the largest part of Europe 293 (all Europe except the Iberian Peninsula, southern Italy, extreme North Europe and some points 294 like Milan or Lyon). It indicates that the results are robust, which may reassure decision-makers. Unfortunately, even if two models give similar results, they can both be wrong. For this reason, a 295 296 diagnosis of good robustness remains difficult to exploit. On the contrary, large differences 297 between the results of two models shows that, at least, one of the models is wrong. In such case, 298 the information provided by the comparison may worry decision-makers but become very valuable for model developers and data providers. Observing the map of figure 6 shows clearly that the 299 Iberian Peninsula and the southern Italy are not well simulated by at least one of the SHERPA 300 301 configurations. This should encourage the developers of CHIMERE and EMEP to control their 302 models and their data in these regions. I advise the authors to insist on this point which seems to me one of the major contributions of their work. 303

Although precise suggestions directly linked to the exact causes of differences between S-EMEP and S-CHIMERE (emissions, meteorology, CTM, SHERPA approximation...) are not possible with the current methodology, we agree that locations where models diverge can be used to trigger further discussion by the model developers. This is indeed one of the contributions of this work and we better stressed this point in the revised version of the paper (in the Conclusions part).

But the evaluation of the difference between two CTM like EMEP and CHIMERE required some 311 wariness. Indeed, SHERPA does not reproduce exactly the results of a CTM generating some 312 errors which will be probably different for EMEP and CHIMERE. The differences which appear 313 between EMEP and CHIMERE will be amplified or damped by SHERPA. So that, high differences 314 between the two SHERPA configurations could hide low differences between EMEP and 315 CHIMERE and vice et versa. This problem has not been commented and is even not mentioned 316 317 in this article. I advise the authors to address this point. I suppose they can easily refer to the SHERPA accuracy that have been estimated in their previous publications. 318

In the revised Supplementary Material, we now included more discussion about the errors attached to the SHERPA approximation. In particular, Figures 4 and 5 show the percentage bias

- errors for different validation scenarios, for the S-CHIMERE and S-EMEP SRR. However, it is not
- 322 possible to extrapolate these average 'percentage bias errors' into specific city errors because these depend on the sector considered, on the area over which emission reductions are applied,etc...

The authors use the Pearson correlation to evaluate the differences between the two SHERPA configurations, which is perhaps not the best statistical indicator. The Pearson coefficient does not spot situations where the results of one of the models are proportional to the other. Let suppose, for example, that the results of one of the models is constantly twice the results of the other model. The Pearson coefficient will then be equal to 1. I advise the author to use another indicator, like the RMSE, it will probably not change their conclusions but should avoid the problem just mentioned.

- The main aim of this work is to assess the policy implications of using a model rather than another.
- This is why we focus on the ranking of the contributions rather than on their absolute values. The ranking is indeed the information that is used to start designing an air quality plans. The Pearson
- 336 coefficient is a good indicator for this purpose whereas the RMSE might give misleading
- information (the example given by the Reviewer would lead to different information while the
- decision would remain unchanged). We now stressed this point in the revised document, at line246.
- 340

Then, it could be interesting to evaluate (even roughly) a threshold above which the differences observed between the two SHERPA configurations reflect significant differences between the two systems of models EMEP and CHIMERE. This would help locate the areas where the differences between EMEP and CHIMERE are proven with near certainty.

We agree with the reviewer. However, it is not possible to evaluate this threshold at this stage.
For doing this, we would need an estimate of the SHERPA uncertainty for each city, sector and
precursor, something we only have for some validation simulations.

351

**Prioritising the sources of pollution in European cities: do air quality modelling applications provide consistent responses?**

Bart Degraeuwe, Enrico Pisoni, Philippe Thunis

European Commission, Joint Research Centre (JRC), Ispra, Italy

Correspondence to: E. Pisoni (enrico.pisoni@ec.europa.eu)

Abstract. To take decisions on how to improve air quality, it is useful to perform a source allocation study that 6 7 identifies the main sources of pollution for the area of interest. Often source allocation is implemented performed with 8 a Chemical Transport Model (CTM) but unfortunately, even if accurate, this technique is time consuming and 9 complex. Comparing the results of different CTMs to assess the uncertainty on the source allocation results is even 10 more difficult. In this work, we compare the source allocation (for PM2.5 yearly averages) on in 150 major cities in 11 Europe, based on the results of two CTMs (CHIMERE and EMEP), approximated through with the SHERPA 12 (Screening for High Emission Reduction Potential on Air) approach. Although contradictory results occur in some 13 instancescities, the source allocation results obtained with the two SHERPA simplified models lead to similar results 14 in most cases, eEven though the two CTMs use different input data and configurations, in most cases the source 15 allocations with the SHERPA simplified models give similar results. But there are also cases where results (in terms 16 of source allocation for PM2.5 yearly averages) are contradictory.

**17 1. Introduction**

Air quality models are useful tools to perform a variety of tasks like assessment (simulating the concentrations fields 19 at a given moment), forecasting (reproducing-predicting future concentrations) and source allocation/planning 20 (evaluating priorities of interventions, and the impact of potential emission reduction policies on concentrations). For 21 assessment (Alvaro Gomez-Losada et al., 2018) and forecasting (Corani et al., 2016), it is possible to compare the 22 model results with observations. For example, FAIRMODE1 (the Forum for air quality modelling in Europe) i.e. 23 provides proposes methodstools as the Model Quality Indicator and Model Quality Objective (Pernigotti el al., 2013b; 24 Viaene et al., 2016) to assess the quality of the model results for a given application., as like the Model Quality 25 Indicator and Model Quality Objective (Pernigotti el al., 2013b; Viaene et al., 2016). However, for source allocation 26 and planning, there is no benchmark against which to compare the model results for source allocation and planning, 27 as. In this context air quality models are simulating no measurements are available to test the impact of theoretical 28 emission reduction scenarios on concentrations, for which no measurements are available. These scenarios are usually 29 implemented considering alternative policy options that might never become real. So, even if they are very useful to

<sup>1 The Forum for Air quality Modeling (FAIRMODE) was launched in 2007 as a joint response initiative of the European Environment Agency (EEA) and the European Commission Joint Research Centre (JRC). The forum is currently chaired by the Joint Research Centre. Its aim is to bring together air quality modelers and users in order to promote and support the harmonized use of models by EU Member States, with emphasis on model application under the European Air Quality Directives. For more details, see https://fairmode.jrc.ec.europa.eu/.

evaluate ex-ante the impact of possible policy options, it is hard to judge the uncertainty quality associated toof these
results. SoOn the other hand,5 the uncertainty on associated tothe source allocation results s given by an air quality
model can be evaluated assessed by comparing it-them with the results of from other -air quality models (Thunis et al., 2007; Cuvelier et al., 2010; Pernigotti et al., 2013). Both the absolute and relative impacts of emission reductions
can then be compared. Even if models disagree about the absolute concentration reductions, they might still identify
the same sources as main contributors to the air pollution in the area of interest. If model results are consistent one
can assume that policies based on these results will be effective.

[revised manuscript text omitted]

The focus of this study is on PM2.5 yearly averages, asbecause this is the pollutant with the highest impact on human health, and is therefore a key focus offor policy makers in Europe. We also stress the fact that bBecause a large number of sources contribute to PM2.5 concentrations at one location, this is also the most challenging pollutant to manage in

**106 air quality plans.**

The paper is structured as follows. We briefly present the two Chemical Transport Model and their set-up in Section

- We then describe the SHERPA methodology and its assumptions in Section 3. Section 4 details the methodology
   followed for the source allocation, while the inter-comparison of the results is presented in Section 5. Conclusions are
- 110 proposed in Section 6.

**111 2. CHIMERE and EMEP Chemical Transport Models: set-up and simulations**

In this work, we used two set of model simulations, performed with two of the leading air quality modelschemical 113 transport models in Europe: CHIMERE and EMEP. More details on the models can be found in Mailler et al., 2017 114 and Couvidaet et al., 2018 (for CHIMERE) and Simpson et al., 2012 (for EMEP). Because aA brute force source 115 allocation for 150 cities with these models would be too time consuming.; instead here we use two sets of SHERPA 116 Source Receptor Relationships (SRR), each based on a training set of about 20 CHIMERE and EMEP CTM 117 simulations to develop a set of SHERPA Source Receptor Relationships (SRR). Theseis SRR set is are then used to 118 perform directly the source allocation. Details on the SHERPA training and validation for CHIMERE can be found in 119 Clappier et al., 2015, and for EMEP in Pisoni et al., 2019.

The CHIMERE and EMEP modelling set-up are different differ in the following aspects: The key differences between
 the two modelling configurations are detailed below:

- Grid setting: CHIMERE uses a grid of 0.125 degrees longitude by 0.0625 degrees latitude, corresponding to rectangular cells of more or less 9 by 7 km (in the centre of the domain) whereas EMEP uses a regular grid of 0.1 by 0.1 degrees, corresponding to rectangular cells of more or less 7 by 11 km.
- Emissions: The CHIMERE emission reference year is 2010 with a gridding based on the EC4MACS project
   proxies (Terrenoire et al., 2015) while EMEP uses a JRC set of emissions (Trombetti et al., 2017) based on
as reference year.
- Boundary conditions: The size of the modelling domains differs. The CHIMERE domain extends from 10.5°
   East to 37.5° West and between 34° and 62° North while the EMEP domain extends from 30° East to 90°
   West and between 30° and 82° North.
- Meteorology: The two models use a different reference meteorological year; 2009 for CHIMERE and 2014
   for EMEP; both meteorological fields are modelled through the Integrated Forecasting System (IFS) of
   ECMWF.
- Model Parameterization: Apart from the vertical and/or horizontal resolutions, transport, deposition, chemical processes might beare reproduced with different levels of complexity in the two models.
- More details on the model simulations and settings can be found in Clappier et al., 2015 and Pisoni et al., 2019. Some
- 137 of the vValidation results for the two model configurations (CHIMERE and EMEP) are briefly presented in the
- 138 Supplementary Material, showing similar performances (for CHIMERE and EMEP) in terms of comparison against observations. For CHIMERE the relation between predictions and observations at background stations is best 140 characterised by a line through the origin with slope of 1.05, indicating a slight under-prediction. The standard error 141 is 5.7 µg/m3 and uniform over the range of concentrations. The R2 is 0.9. Concentrations at traffic and industrial 142 stations are underestimated by roughly 10%. For EMEP the relation between predictions and observations is best 143 characterised by a power low with exponent 0.66. The data show a relative standard error constant over the range of 144 concentrations and equal to 30%. Concentrations at traffic stations are under-predicted by 9% and over-predicted at 145 industrial sites by 7%. It is important to note that the use of different input and model set-up (as listed before) represents 146 the usual practice when air quality models are used, at the local scale, to assess the impact of air quality plans. This is 147 why it is important (in this manuscript) here to analyse how this choice influences the results and the subsequent design 148 of an air quality plan; an issue that is often not tackled in the scientific literature. Some differences in results might be 149 due to trends in emissions and concentrations between 2010 and 2014. During this period, concentrations in Airbase 150 stations decrease yearly by 2.2% on average ( $\sigma = 2.7\%$ /year). Hence, only differences larger than about 10% in source 151 apportionment should be considered as significant. 
[revised manuscript text omitted]
 various types 267 ofrelative potentials defined in terms of considered aggregationsperformed scenarios (considering emission 268 reductions for the selected geographical area, for the chosen sector, or for their combinations of geographical areas 269 sectors, ...) and their corresponding correlations, for the same cities.

As said, we present results for 4 cities (Liege, Genova, Turin and Madrid) selected as representative of the different 271

behaviours identified in our analysis.

For Liege (Belgium), the overall (all individual sectors, precursors and areas included, i.e. about 15000 relative 273 potentials) Pearson correlation-3 between the relative potentials of both models SRR is the highest among the 150 274 cities (r=0.99, see Figure 1). Both models identify ammonia emissions from agriculture, outside Belgium, as the main 275 contributor to local PM2.5 concentrations. Primary PM from local industry comes second and NOx from international 276 traffic third. Although the lower ranked combinations are not identical, they are quite similar. From a policy 277 perspective, the fact that both modelling applicationsSRR provide similar information is a sign of robustness. It 278 increases our confidence in the priority of interventions (which sectors-areas to act at first to achieve the maximum 279 air quality improvement)-proposed by each model.. The values of for the the different main sector-precursor-areas 280 contributions (expressed as relative potentials) are reported in Table 1.

Table 1: Top 10 area-sector-precursor combinations contributirelative potentialsng to the PM2.5 concentrations in Liege (B).

| area          | sector         | precursor | emep_rp | emep.rank | chimere_rp | chimere.rank |
|---------------|----------------|-----------|---------|-----------|------------|--------------|
| International | Agriculture    | NH3       | 22.9    | 1         | 20.6       | 1            |
| FUA           | Industry       | PPM       | 12.6    | 2         | 12.4       | 2            |
| International | Road Transport | NOx       | 7.5     | 3         | 6.9        | 3            |
| International | Industry       | NOx       | 4.9     | 5         | 5.2        | 4            |
| National      | Agriculture    | NH3       | 4.2     | 6         | 4.6        | 5            |
| International | Industry       | SOx       | 5.1     | 4         | 2.3        | 10           |
| International | Residential    | PPM       | 2.2     | 7         | 2.5        | 8            |
| FUA           | Road Transport | PPM       | 2.1     | 10        | 2.9        | 6            |
| International | Industry       | PPM       | 2.2     | 8         | 2.4        | 9            |
| FUA           | Industry       | SOx       | 1.9     | 15        | 2.7        | 7            |
| International | Other          | NOx       | 2.2     | 9         | 1.9        | 13           |

A breakdown analysis for Liege is proposed in Figure 1 where correlations are expressed-calculated for different data
 relative potentials that are aggregated in terms of sectors (aggregations. In addition to the overall correlation (75000 values), values are also proposed for data grouped by sectors (150 cities x-5 relative potentialssectors), by-area (150 cities x-4 areasrelative potentials) or by area/sectors (150 cities x-5 pr3ecursors x 5 pollutantsrelative potentials). In the case of Liege, all correlations are consistently very good.

The main aim of this work is to assess the policy implications (i.e. which source to tackle first) of using a model rather than another. This is why we focus on the ranking of the contributions (Pearson correlation) rather than on their absolute values; that means, this is way we use in this context the Pearson correlation.

Figure 1: Correlation between S-EMEP and S-CHIMERE relative potentials <del>relative potentials of S-EMEP and S-CHIMERE</del> for different sector-area-precursor source aggregations in Liege (B).

Unfortunately, the agreement is not always as so good. For the city of Genova (Table 2 and Figure 2), both models agree that national/international ammonia emissions from agriculture areas are the largest contributor to local PM2.5 (see Table 2). But the third position in the priority ranking is occupied by NOx from national traffic for S-EMEP while it is PPM from the national residential sector for S-CHIMERE. However, the overall correlation still reaches 89% and the absolute values of the third ranked sectors are quite closetwo main sources are similar. The agreement between the two models is therefore still satisfactory. It is interesting to note that for relative potentials area-aggregated -relative potentialsaggregated per area, the correlation drops to 42%, pointing highlighting possible to-differences in the way emission inventories are spatially distributedion of in the two models the two emission inventories.

| R             | elative Potentials |           |         |           |            |              |
|---------------|--------------------|-----------|---------|-----------|------------|--------------|
| area          | sector             | precursor | emep_rp | emep.rank | chimere_rp | chimere.rank |
| National      | Agriculture        | NH3       | 14.5    | 1         | 11.3       | 1            |
| International | Agriculture        | NH3       | 6.8     | 2         | 10.1       | 2            |
| National      | Residential        | PPM       | 4.3     | 4         | 4.7        | 3            |
| FUA           | Residential        | PPM       | 3.2     | 5         | 3.5        | 4            |
| National      | Road Transport     | NOx       | 4.9     | 3         | 2.6        | 8            |
| FUA           | Road Transport     | NOx       | 3.2     | 6         | 2.8        | 7            |
| International | Industry           | SOx       | 2.2     | 10        | 3.4        | 5            |
| National      | Industry           | SOx       | 1.7     | 15        | 2.5        | 9            |
| International | Residential        | PPM       | 1.4     | 18        | 2.8        | 6            |
| FUA           | Road Transport     | PPM       | 1.4     | 17        | 2.1        | 10           |
| FUA           | Other              | NOx       | 2.5     | 8         | 0.7        | 21           |
| FUA           | Industry           | NOx       | 2.4     | 9         | 0.0        | 59           |
| FUA           | Industry           | SOx       | 3.1     | 7         | 0.0        | 62           |

 Table 2: Top 10 area-sector-precursor combinations contributirelative potentials ng to the PM2.5 concentrations in Genova (IT).